# Calculated Maximal Volume Ventilation (cMVV) as a Marker of Early Respiratory Failure in Amyotrophic Lateral Sclerosis (ALS)

**DOI:** 10.3390/brainsci14020157

**Published:** 2024-02-03

**Authors:** Umberto Manera, Maria Claudia Torrieri, Cristina Moglia, Antonio Canosa, Rosario Vasta, Francesca Palumbo, Enrico Matteoni, Sara Cabras, Maurizio Grassano, Alessandro Bombaci, Alessio Mattei, Michela Bellocchia, Giuseppe Tabbia, Fulvia Ribolla, Adriano Chiò, Andrea Calvo

**Affiliations:** 1Umberto Manera, ALS Centre, ‘Rita Levi Montalcini’ Department of Neuroscience, University of Turin, Via Cherasco 15, 10126 Turin, Italyandrea.calvo@unito.it (A.C.); 2SC Neurologia 1U, AOU Città della Salute e della Scienza di Torino, 10126 Turin, Italy; 3Academic Neurology Unit, San Luigi Gonzaga University Hospital, 10136 Orbassano, Italy; 4Institute of Cognitive Sciences and Technologies, Consiglio Nazionale delle Ricerche C.N.R., 00185 Rome, Italy; 5S.C. Pneumologia, S. Croce and Carle Hospital, 12100 Cuneo, Italy; 6S.C. Pneumologia U, AOU Città della Salute e della Scienza di Torino, 10126 Turin, Italy

**Keywords:** amyotrophic lateral sclerosis, pulmonary function tests, maximal volume ventilation, forced vital capacity, spirometry

## Abstract

Respiratory failure assessment is among the most debatable research topics in amyotrophic lateral sclerosis (ALS) clinical research due to the wide heterogeneity of its presentation. Among the different pulmonary function tests (PFTs), maximal voluntary ventilation (MVV) has shown potential utility as a diagnostic and monitoring marker, able to capture early respiratory modification in neuromuscular disorders. In the present study, we explored calculated MVV (cMVV) as a prognostic biomarker in a center-based, retrospective ALS population belonging to the Piemonte and Valle d’Aosta registry for ALS (PARALS). A Spearman’s correlation analysis with clinical data and PFTs showed a good correlation of cMVV with forced vital capacity (FVC) and a moderate correlation with some other features such as bulbar involvement, ALSFRS-R total score, blood oxygen (pO_2_), carbonate (HCO_3_^−^), and base excess (BE), measured with arterial blood gas analysis. Both the Cox proportional hazard models for survival and the time to non-invasive ventilation (NIV) measurement highlighted that cMVV at diagnosis (considering cMVV(40) ≥ 80) is able to stratify patients across different risk levels for death/tracheostomy and NIV indication, especially considering patients with FVC% ≥ 80. In conclusion, cMVV is a useful marker of early respiratory failure in ALS, and is easily derivable from standard PFTs, especially in asymptomatic ALS patients with normal FVC measures.

## 1. Introduction

Amyotrophic lateral sclerosis (ALS) is the most common form of motor neuron disease (MND), a family of neurodegenerative diseases characterized by the progressive loss of bulbar, limbs, and respiratory muscles [1]. Among the different body regions involved in the degeneration of motor neurons, the thoracic/respiratory muscles are often considered to be one of the last types to become involved [2], and only 1% of patients have a respiratory onset phenotype [3].

The evaluation of early respiratory failure is one of the more elaborate challenges in ALS patient management for many reasons. Firstly, as for most chronically progressive conditions, patients present different degrees of adaptation to hypoventilation, and the relationship between respiratory test alterations and respiratory symptoms is widely unclear [4]. Secondly, consulting the most updated international guidelines on mechanical ventilation [5,6,7,8], there is no international consensus on the best timing for non-invasive ventilation (NIV) adaptation. Thirdly, although forced vital capacity (FVC) is the most commonly used pulmonary function test and outcome measure in clinics, many other respiratory measures, such as nocturnal oximetry [9], capnography [10], sniff nasal inspiratory pressure (SNIP) [11], and arterial blood gas analysis (ABGs) [12,13] have shown high reliability in detecting early signs of respiratory failure.

The maximal voluntary ventilation (MVV) is defined as the maximum volume of air that a subject can breathe over a specified period of time (12 s for normal subjects), expressed in L/min [14]. By definition, MVV measures respiratory muscle endurance rather than strength and may be more sensitive than FVC in predicting early respiratory deterioration, considering that none of the above mentioned respiratory measures is specifically designed for respiratory fatigue assessment. Due to the heterogeneity of ALS, utilizing this measure in addition to tests that assess other factors, such as total vital capacity, nocturnal hypoventilation, daytime hypercapnia, and inspiratory muscle weakness, should be useful.

MVV is not generally included in the set of lung function tests necessary for diagnosis or follow-up of primary lung diseases due to its good correlation with the forced expiratory ventilation in the first second test (FEV1); however, its role in the assessment of some conditions, such as neuromuscular disorders, has been reported [15,16]. In clinical practice, a calculated MVV (cMVV) is usually estimated by multiplying FEV1 by a constant value; measured MVV (mMVV) and cMVV, when compared in different populations, such as patients with chronic obstructive pulmonary disease (COPD) or disability claimants [17,18,19], are often discordant. There is no consensus in scientific literature on which measure is more reliable and useful for patient management. In neuromuscular conditions, such as myotonic muscular dystrophy type 1 (DM1), both mMVV and cMVV have been shown to be correlated with hypercapnia, being the first PFTs to decrease in hypercapnia as compared to FVC and FEV1 [16].

Concerning ALS, a recent paper suggested MVV as a potentially useful diagnostic and monitoring marker, as it is able to capture subtle early respiratory modifications and is correlated with both PFTs and neurophysiological studies of respiratory muscles [20]. No study of cMVV in ALS has been performed yet, necessitating exploration of the correlation of cMVV with other respiratory measurements, as well as its prognostic role.

The aim of our study is to provide an in-depth analysis on these topics and on the ability of cMVV to differentiate patients with early respiratory involvement from patients with normal respiratory function in a retrospective, center-based cohort of ALS patients. Based on the retrospective nature of our study, comparing mMVV and cMVV was beyond the scope of our study.

## 2. Materials and Methods

### 2.1. Descriptive Statistics

We obtained data from the Piemonte and Valle D’Aosta ALS register (PARALS). We included all patients who were diagnosed with ALS in the 1995–2015 period in the Turin ALS Centre and who underwent regular spirometry during the diagnostic workup. Methods relating to the PARALS register have been exhaustively described elsewhere [3]. We collected pulmonary function tests (PFTs) performed by ALS patients within 4 months from diagnosis, associating each PFT with the nearest ALSFRS-R scale performed.

We also evaluated, in a separate analysis, the data of the first ABGs and PFTs performed simultaneously during the respiratory workup, using another dataset already published [12]. This dataset also reported the nearest ALSFRS-R performed, and we derived from it data on respiratory symptoms (by using ALSFRS-R item 10 and 11) and bulbar dysfunction, defining bulbar involvement when the sum of items 1, 2, and 3 resulted in a value < 12 [2]. Patients with severe pulmonary, metabolic, and kidney diseases and those with signs of uncompensated acidosis/alkalosis (ABGs pH < 7.35 and > 7.45) were excluded. Specifically, we considered “severe pulmonary disease” as the presence of: (1) severe ongoing asthma documented by the persistent use of bronchodilator/corticosteroid drugs; (2) severe ongoing COPD documented by the persistent use of bronchodilator/corticosteroid drugs; (3) lung tumor (both historical and ongoing); (4) documented primary lung diseases (pulmonary fibrosis, sarcoidosis, etc., both historical and ongoing); and (5) active, ongoing pulmonary infectious disease (namely pneumonia). Age at onset, sex, site of onset, date of diagnosis, death/tracheostomy status, date of ABG/spirometry, and date of ALSFRS-r scale were collected for each patient. Survival was calculated from diagnosis to the day of death/tracheostomy, or to 31 December 2018, and progression rate at diagnosis (ΔALSFRS-R) was reported as the ratio between the difference 48—ALSFRS-R at diagnosis, and the time interval in months between onset and diagnosis.

### 2.2. Calculated MVV Definition

Following a review of the literature [19,21,22], which involved evaluating different proposed equations and selecting those used in healthy adults or patients with common pulmonary conditions, we decided to use the below equation for calculated MVV (cMVV). This gives the so-called cMVV(40), derived from measured forced expiratory ventilation in the first second (FEV1), through the following formula:cMVV(40) = FEV1 (L) ∗ 40

Other methods were discarded to simplify the interpretation of the results, and also because they were simply a transformation of cMVV(40). An example of this is the MVV(35) = FEV1 (L) ∗ 35 equation. Some other methods were discarded as they had been designed for specific populations [17,23].

To validate cMVV performance, we applied another method, proposed by Dillard & colleagues [18]. This method also takes into account the maximum inspiratory flow rate (MIFR) or peak inspiratory flow (PIF) to adjust cMVV calculation according to the following formula:cMVV(Dillard) = 30.77 ∗ FEV1 (L) + 5.94 ∗ PIF (L/s) − 4.77

Considering that cMVV(40) is based on absolute FEV1 (which does not account for anthropometric measures), and not on FEV1%, we also explored the prognostic ability of FEV1%, as well as the FEV1/FVC ratio (modified Tiffeneau-Pinelli index), to better understand the correlation between cMVV(40) and obstruction signs.

### 2.3. Statistical Analysis

We analyzed differences in discrete and continuous variables using the χ^2^ test and Student’s *t* test, or the Kruskal-Wallis and Mann-Whitney U tests, respectively.

The correlations between cMVV(40), FVC, and other clinical variables were calculated using the non-parametric two-tailed Spearman’s rank correlation test.

For time-to-event analysis, we derived Kaplan-Meier curves using log-rank tests and utilized Cox proportional hazard models which were adjusted for several well-determined prognostic factors. We evaluated the effect of cMVV(40), cMVV(Dillard), and FEV1% on overall survival and time to non-invasive ventilation (NIV) start. This was performed separately from evaluations regarding diagnosis to death/tracheostomy or NIV adaptation, using both data collected at diagnosis or at the second time point during pulmonary workup. To specifically study the additional value of cMVV(40), cMVV(Dillard), and FEV1% compared to FVC, we stratified time-to-event analysis according to the value of normality generally considered for FVC (><80%) and using quartiles (rounded) for cMVV(40). A *p* value < 0.05 was considered significant. In order to better define the best cut-off values for the cMVV(40), ROC curves were computed so as to study the sensitivity and specificity of each cut-off for survival at 3, 6, and 12 months, and NIMV start within 6 months from PFT. The Youden Index (computed as sensitivity + specificity − 1) was then calculated.

Data were analyzed using IBM SPSS Statistics for Windows, Version 28.0.1.0 (IBM Corp. Released 2021). 

## 3. Results

### 3.1. Descriptive Statistics

According to the inclusion criteria, from the overall registry cohort (N = 2840), 1342 patients (47.3%) underwent spirometry and performed PFTs during diagnostic workup as part of the Turin ALS Center cohort. Among them, 1287 eligible ALS patients (95.9%) were included in the analysis. Descriptive statistics for the included cohort are summarized in Table 1.

The subgroup of patients (N = 576) for which PIF was available was compared to the whole cohort, showing no significant differences in the main variables.

For 484 of these patients, we collected a second spirometry reading, with complete FEV1 performed as part of the respiratory assessment, together with arterial blood gas (ABGs) analysis [12]. The results for this subgroup of patients were significantly different from the total cohort in terms of time from disease onset, total ALSFRS-R score, FVC%, FEV1%, and cMVV(40). This was expected, considering that in our retrospective dataset, the respiratory assessment was generally performed later in the disease course. Despite this, no significant selection bias was detected, according to all other clinical variables.

### 3.2. Correlations with PFTs, ABGs, ALSFRS-R and Other Clinical Features

At diagnosis, cMVV(40) significantly correlated with FVC% (0.626, *p* < 0.001), FEV1% (0.669, *p* < 0.001), age at onset (−0.426, *p* < 0.001), age at PFT (−0.429, *p* < 0.001), ALSFRS-R total score (0.319, *p* < 0.001), and with bulbar involvement (0.356, *p* < 0.001). The cMVV(40) showed only low correlation with ΔALSFRS-R (−0.209, *p* < 0.001), onset-PFT interval (−0.072, *p* < 0.001), and specific ALSFRS-R respiratory items (item 10: 0.214, *p* < 0.001; item 11: 0.176, *p* < 0.001: item 12: 0.116, *p* < 0.001). No significant correlation was found between cMVV(40) and FEV1/FVC ratio (0.033, *p* = 0.237). Included in the Appendix A of this paper are two figures that better explain the distribution of patients according to FVC%, FEV1%, and cMVV(40) values.

cMVV(Dillard) and cMVV(40) were highly correlated (0.983, *p* < 0.001) in the subset of patients with PIF available. cMVV(Dillard) results were minimally less correlated with FVC% (0.584, *p* < 0.001) and FEV1% (0.611, *p* < 0.001) compared to cMVV(40) results. Interestingly, PIF results were more correlated with both cMVV(40) (0.693, *p* < 0.001) and cMVV(Dillard) (0.807, *p* < 0.001) results than FVC% and FEV% results. No significant correlation was found between the FEV1/FVC ratio and the cMVV(Dillard) (0.028, *p* = 0.504). Peak expiratory flow (PEF) results were highly correlated with both cMVV(40) (0.823, *p* < 0.001, N = 1115 patients) and cMVV(Dillard) (0.851, *p* < 0.001, N = 576 patients) results.

cMVV(40) results also moderately correlated with ABGs parameters, such as pCO_2_ (−0.278, *p* < 0.001), HCO_3_^−^ (−0.323, *p* < 0.001), BE (−0.318, *p* < 0.001), and pO_2_ (0.350, *p* < 0.001). In spinal patients, cMVV(40) was slightly higher in patients with upper limbs onset, in comparison to patients with lower limbs onset (90.8, IQR 66.4–116.9 vs. 83.2 IQR 64.4–106.4, *p* = 0.012). This difference may be related to the disease duration from onset, considering that it was significantly shorter in patients with upper limbs onset (11 months, IQR 7–19 months vs. 13 months, IQR 8–24 months, *p* > 0.001).

### 3.3. Time-to-Event and ROC Analysis: Overall Survival and Time-to-NIV

We applied different Cox proportional hazard models, considering overall survival and time to non-invasive ventilation (NIV) start (see Table 2).

Cox proportional hazard models adjusted for sex, age, progression rate, and site of onset confirmed that HRs for both survival and time to NIV decreased significantly with an increase in cMVV(40), both in univariate and multivariate analysis with continuous and categorized values. To understand the possible role of bulbar involvement in PFT performance, we stratified the analysis for bulbar involvement, but it did not significantly change the hazard ratios. In Cox models adjusted also for FVC%, cMVV(40) was found to have no significant association with survival. To better comprehend the interplay between these two indexes, we stratified the analysis according to the FVC% limit of normality (80%); in patients with normal FVC (≥80), cMVV(40) was significantly associated with overall survival. The FEV1/FVC ratio was found to not be significantly associated with survival when examined under univariate analysis (see Table 2), while FEV1% was found to be significantly associated with survival when examined under univariate and multivariate analysis. However, after stratification for FVC values, this significant association ceased to exist (see Appendix A).

We performed a Kaplan-Meier analysis (Figure 1). We subdivided patients according to FVC% and cMVV(40) values and maintained 80 as the cut-off for both measures in order to simplify memorization of the cut-off in clinical practice, but also considering that it was not significantly distant to the median cMVV(40) value of our entire cohort.

Kaplan-Meier analysis confirmed that patient stratification for cMVV(40), especially in patients with FVC% ≥ 80%, enabled the identification of two different cohorts with different survival rates and non-invasive ventilation timing. In the Appendix A for this paper, we have included the adjusted Cox proportional hazard models (Appendix A) and a Kaplan-Meier analysis (Appendix A) performed using other cMVV(40) cut-offs (≤60; 60–80; ≥80). These can be used alone for patient prognostic stratification.

ROC analysis was performed for cMVV(40) values considering patient survival at 3, 6, and 12 months and NIV start at 6 months. The results are included in the Appendix A. The AUC result was >0.6 for all of the curves. Youden index values for survival at 1 year and for NIV at 6 months confirmed that values of cMVV(40) around the adjusted median (80) can be considered as a valid cut-off in our ALS population.

## 4. Discussion

Maximum voluntary ventilation (MVV) is a pulmonary function parameter that has not been commonly evaluated in the literature. It assesses the maximum amount of air a person can inhale and exhale voluntarily in a given period of time. MVV is particularly interesting in neuromuscular disorders because it provides information on respiratory muscle mechanics and endurance, which are involved in the mechanism of dyspnea and exercise limitation [24]. In our study, we observed that cMVV, a derived measure simply obtainable both from retrospective datasets and in clinical practice using a standard spirometry test using measured FEV1 and PIF, can stratify patients into different prognostic classes.

According to our results, cMVV proved its usefulness as a new respiratory evaluation measure, especially in ALS patients with normal FVC% (≥80%): a rapid calculation of both measures, using the same cut-off of 80, allowed us to distinguish two different populations of patients with a median difference of survival of more than 6 months.

In our study we observed that in ALS patients cMVV and FEV1% measured different aspects of respiratory function compared to what they measure in healthy populations and patients with different clinical conditions. In ALS patients, both cMVV(40) and cMVV(Dillard) were correlated with PIF, a underestimated measurement of inspiratory function, while no significant correlation was found with FEV1/FVC ratio, which is considered the gold standard for obstructive lung disease diagnosis. As reported in scientific literature on ALS, measurements of inspiratory pressures (maximal inspiratory pressure, or MIP and SNIP) are considered more sensitive than FVC in the detection of early respiratory involvement in early MND [25,26,27,28]. Unfortunately, the role of PIF is underestimated in ALS/MND scientific literature and no scientific papers exist regarding this topic. We also pointed out a strong correlation between cMVV and PEF, which have been recently described as useful respiratory biomarkers in ALS patients, especially for home monitoring [29,30]. According to our data, cMVV, PIF, and PEF are suitable for use in clinical practice in order to reveal early respiratory impairments in ALS patients, but further prospective studies are needed.

In other ALS cohorts, MVV has already been shown to have good correlations with other PFTs, such as FVC, slow vital capacity (SVC), FEV1, SNIP, phrenic nerve amplitude, peak expiratory flow (PEF), and total ALSFRS-R [20]. Recent studies examining independent cohorts [12,13,31] have agreed upon the role of blood carbonate (HCO_3_^−^) and base excess (BE) in evaluating early respiratory failure phenomena such as nocturnal hypoventilation, and we also observed a moderate correlation of cMVV with these two ABGs measures.

The cMVV also moderately correlates with pO_2_ and age in our population; this is not surprising, considering that this measure reflects also some features of aging lungs that are obviously age-dependent and explain the correlation with a reduced blood oxygen absorption.

As is already known for FVC [32,33], cMVV also showed poor correlation with respiratory symptoms evaluated by ALSFRS-R scale respiratory items. Respiratory failure initially presents in ALS without significant daytime symptoms, like dyspnea or orthopnea investigated by ALSFRS-R [34]. New ALS specific respiratory questionnaires have been developed to overcome ALSFRS-R limitations, including the evaluation of early nocturnal and daytime symptoms [35].

In smaller ALS cohorts with longitudinal evaluations, MVV has been shown to undergo a progressive reduction alongside disease progression, as expected [36]. This was evident also in our results, considering the median difference of almost 10 points in the cohort with ABGs, whose median disease duration was 2.5 months longer than that of the cohort as a whole.

In many papers on the differences between measured MVV (mMVV) and calculated MVV (cMVV), which is obtained by multiplying the MVV with measured FEV1, a standardized and reproducible maneuver, with an appropriate factor (generally 35 or 40), has been observed to be significant [16,17,19,21,22,23,37]. Preferences regarding the use of cMVV versus mMVV for ventilator capacity calculation are dependent on the studied cohort: for example, in pediatric populations, cMVV has been shown to be a more accurate surrogate measure of maximum ventilator capacity, most likely due to inadequate effort and inability to perform MVV maneuvers [38]. Cognitive and behavioral impairment can affect the assessment of ventilatory function in ALS [39], and for this reason, simpler additive information, obtainable by standard PFT, could help to improve patient management.

This is the first study evaluating the prognostic role of cMVV in a well-powered ALS cohort. We confirmed the prognostic reliability of cMVV, reflecting previous studies that have previously been conducted using mMVV [20,36] as a prognostic marker. The retrospective nature of the study using a center-based population was a limitation that precluded us from directly comparing cMVV with mMVV, MEP, MIP, or SNIP. Before recommending the use of cMVV in clinical practice, further prospective multi-center studies conducted in different ALS populations and during different stages of disease progression are needed to better evaluate the relationship of cMVV with other available respiratory parameters, mainly nocturnal oximetry, MIP/MEP, and SNIP. Nevertheless, our retrospective data suggests that cMVV measured at diagnosis provides a prognostic stratification of ALS patients. Moreover, its correlation with other validated respiratory tests proves that cMVV is a good surrogate of early ventilatory dysfunction related to ALS.

## 5. Conclusions

In conclusion, our data on cMVV confirmed its reliability as a functional respiratory measure able to identify patients with early respiratory failure. Being easily derivable from standard PFTs, we highlighted its potential additive role in prognostic stratification, especially in asymptomatic ALS patients with normal PFT results. Our results can orient clinicians and researchers toward a more nuanced assessment of respiratory function, targeting and anticipating interventions to enhance quality of life and improve therapeutic adherence to non-invasive ventilation, by determining early recognition of asymptomatic patients with underlying respiratory conditions, or who are prone to developing respiratory complications in the near future.

## Figures and Tables

**Figure 1 brainsci-14-00157-f001:**
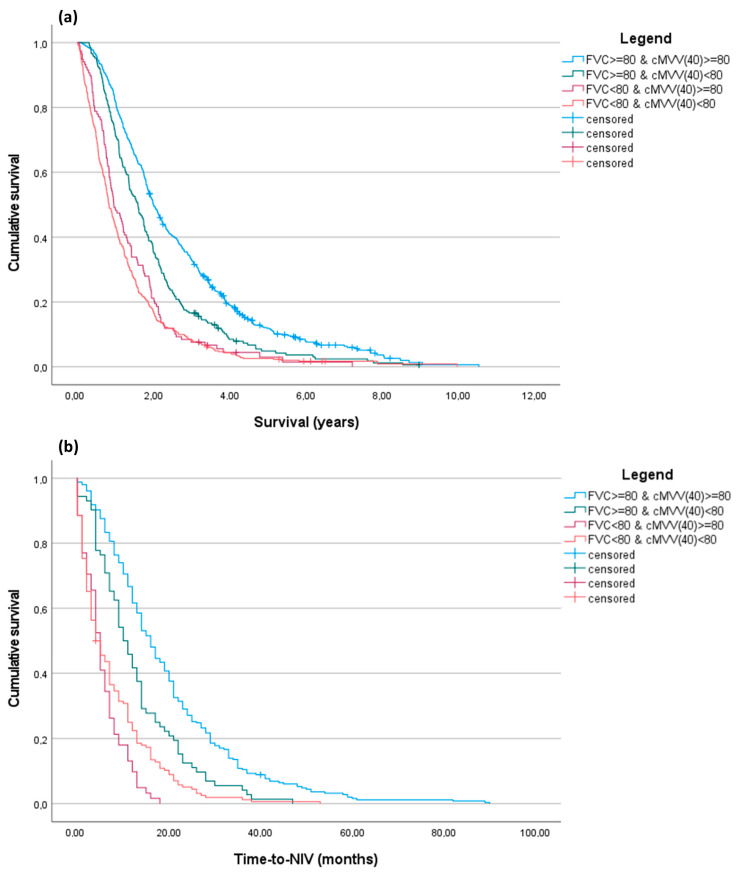
Kaplan-Meier curves for overall survival and time-to-NIV. Patients were subdivided into four categories according to the combination of FVC% values and cMVV(40), using for both a cut-off of 80. (**a**) Overall survival (years) from PFT performance. The result of the pairwise log-rank test *p* was <0.001 for all combinations, except for the FVC% < 80 groups (*p* = 0.064); (**b**) Time-to-NIV (months) from PFT performance. The result of the pairwise log-rank test *p* was <0.001 for all combinations, except for the FVC% < 80 groups (*p* = 0.079).

**Table 1 brainsci-14-00157-t001:** Descriptive statistics.

	Whole Cohort (N = 1287)	Cohort with PIF (N = 576)	Cohort with ABGs (N = 484)	Whole vs. PIF Cohort	Whole vs. ABGs Cohort
	Median (IQR)	Median (IQR)	Median (IQR)	*p* *	*p* *
Age at onset (years)	66.2 (58.9–72.7)	65.9 (57.9–72.5)	66.1 (58.8–73.5)	0.520	0.588
Age at PFT performance (years)	67.4 (60.0–73.8)	67.2 (59.5–73.6)	67.6 (60.1–73.5)	0.487	0.941
Onset-PFT interval (months)	11.0 (6.0–18.0)	11.0 (7.0–18.0)	13.5 (8.6–22.2)	0.893	**<0.001**
ALSFRS-R total score	42.0 (37.0–45.0)	42.0 (37.0–45.0)	40.0 (34.0–44.0)	0.968	**<0.001**
ΔALSFRS-R	0.50 (0.29–1.00)	0.50 (0.30–0.90)	0.55 (0.33–1.00)	0.774	0.137
FVC%	85.9 (66.9–101.2)	88.7 (70.6–101.9)	75.6 (58.5–92.0)	0.069	**<0.001**
FEV1%	86.4 (68.1–102.7)	89.6 (71.5–102.7)	77.2 (58.4–95.5)	0.106	**<0.001**
FEV1/FVC ratio	81.8 (75.9–87.7)	81.1 (75.6–86.2)	-	0.054	-
cMVV(40)	81.6 (59.6–104.8)	84.0 (63.3–109.4)	72.2 (53.7–94.9)	0.084	**<0.001**
PEF (L/s) (N = 1115)	4.3 (3.0–5.7)	4.3 (3.1–5.7)	-	0.984	**-**
PIF (L/s)	-	2.6 (1.7–3.6)	-	-	-
cMVV(Dillard)	-	85.7 (64.1–110.1)	-	-	-
pH	-	-	7.43 (7.41–7.44)	-	-
pO_2_ (mmHg)	-	-	81.0 (73.8–87.0)	-	-
pCO_2_ (mmHg)	-	-	40.0 (34.0–44.0)	-	-
BE (mEq/L)	-	-	1.38 (−0.20–3.12)	-	-
HCO_3_^−^ (mmol/L)	-	-	25.5 (23.7–27.3)		-
	N (%)	N (%)	N (%)	*p* ^§^	*p* ^§^
Sex					
Female	587 (45.6)	253 (43.9)	217 (44.8)	0.532	0.792
Male	700 (54.4)	323 (56.1)	267 (55.2)		
Site of onset					
Bulbar onset	427 (33.2)	184 (31.9)	162 (33.5)	0.872	0.912
Limbs onset	849 (66.0)	387 (67.2)	316 (65.3)		
Respiratory onset	11 (0.9)	5 (0.9)	6 (1.2)		
Total	1287 (100.0)	576 (100.0)	484 (100.0)		

PIF: peak inspiratory flow; ABGs: arterial blood gas analysis; * Mann-Whitney U test; ^§^ Chi-square test; all significant results (*p* < 0.05) are written in bold.

**Table 2 brainsci-14-00157-t002:** Cox proportional hazard models.

Univariate Analysis—Overall Survival					
	HR (95% CI)	*p*		HR (95% CI)	*p*
Age at PFT (continuous)	1.022 (1.016–1.028)	**<0.001**	FVC%	0.983 (0.980–0.985)	**<0.001**
Sex	1.132 (1.010–1.269)	**0.033**	FEV1%	0.985 (0.982–0.987)	**<0.001**
Site of onset (B/S)	1.308 (1.166–1.467)	**<0.001**	cMVV(40) (continuous)	0.990 (0.988–0.992)	**<0.001**
ΔALSFRS	1.231 (1.176–1.289)	**<0.001**	FEV1/FVC ratio	1.006 (1.000–1.012)	0.067
Bulbar involvement	0.699 (0.618–0.791)	**<0.001**	PIF (continuous, N = 576)	0.832 (0.777–0.891)	**<0.001**
			cMVV(Dillard) (continuous, N = 576)	0.991 (0.988–0.994)	**<0.001**
cMVV(40) (median adj)			cMVV(40) (quartiles adj)		
<80	1		<60	1	
≥80	0.595 (0.531–0.667)	**<0.001**	60–80	0.611 (0.520–0.717)	**<0.001**
			80–105	0.489 (0.417–0.573)	**<0.001**
			≥105	0.428 (0.364–0.502)	**<0.001**
cMVV(Dillard) (median adj)			cMVV(Dillard) (quartiles adj)		
<85	1		<65	1	
≥85	0.677 (0.572–0.802)	**<0.001**	65–85	0.611 (0.520–0.717)	**<0.001**
<80	1		85–105	0.489 (0.417–0.573)	**<0.001**
≥80	0.617 (0.520–0.731)	**<0.001**	≥105	0.428 (0.364–0.502)	**<0.001**
Multivariate analysis—Overall survival—cMVV(40)					
	Adjustments			HR (95% CI)	*p*
	Age at onset (cont), Sex, Site of onset (B/S), ΔALSFRS (N = 1287)		cMVV(40) (continuous)	0.988 (0.986–0.990)	**<0.001**
			cMVV(40) (median adj)		
			<80	1	
			≥80	0.598 (0.517–0.691)	**<0.001**
			cMVV(40) (quartiles adj)		
			<60	1	
			60–80	0.633 (0.531–0.755)	**<0.001**
			80–105	0.490 (0.409–0.587)	**<0.001**
			≥105	0.404 (0.326–0.500)	**<0.001**
	Age at onset (cont), Sex, Site of onset (B/S), ΔALSFRS, bulbar involvement (N = 1083)		cMVV(40) (continuous)	0.989 (0.986–0.991)	**<0.001**
Stratified analysis—Overall survival—cMVV(40)					
FVC ≥ 80	Age at onset (cont), Sex, Site of onset (B/S), ΔALSFRS (N = 1287)		cMVV(40) (median adj)		
			<80	1	
			≥80	0.785 (0.630–0.979)	**0.032**
FVC < 80	Age at onset (cont), Sex, Site of onset (B/S), ΔALSFRS (N = 1287)		cMVV(40) (median adj)		
			<80	1	
			≥80	0.931 (0.718–1.206)	0.586
Multivariate analysis—time to NIV—cMVV(40)					
	Adjustments			HR (95% CI)	*p*
	Age at onset (cont), Sex, Site of onset (B/S), ΔALSFRS (N = 1287)		cMVV(40) (continuous)	0.983 (0.980–0.987)	**<0.001**
			cMVV(40) (median adj)		
			<80	1	
			≥80	0.525 (0.417–0.660)	**<0.001**
			cMVV(40) (quartiles adj)		
			<60	1	
			60–80	0.626 (0.472–0.830)	**0.001**
			80–105	0.482 (0.360–0.645)	**<0.001**
			≥105	0.278 (0.200–0.388)	**<0.001**

All significant results (*p* < 0.05) are written in bold.

## Data Availability

The data are available upon reasonable request. The data are not publicly available due to privacy and ethical restrictions.

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
