# Peer review of "Calculated Maximal Volume Ventilation (cMVV) as a Marker of Early Respiratory Failure in Amyotrophic Lateral Sclerosis (ALS)"

_brainsci, 2024, doi:10.3390/brainsci14020157_

Round 1
Reviewer 1 Report
Comments and Suggestions for Authors
This is a well written original article that introduces calculated maximal voluntary ventilation (cMVV) as an additional measure of respiratory function in patients with ALS (PALS). The authors postulate that cMVV might help to detect early respiratory failure and predicts time to mechanical ventilation and survival. The main hypothesis of this work is interesting, the sample size is reasonably large, and the statistical approach is sound. However, I have major concerns regarding the very beginning of the methodology (i. e., the choice of cMVV(40) as the parameter of interest) and the very end of data interpretation, i. e. theoretical deductions and the clinical relevance of this study's results.
1.
Firstly, the authors should point out briefly but more clearly why they preferred calculated MVV over measured MVV in PALS already in the introduction. In the methods section, the actual choice for cMVV(40) is not referenced which I find astonishing as this is crucial for the entire manuscript.
Reference 18 and several other publications in the literature report that cMVV(40) or cMVV(35) poorly correlate with cMVV values that are derived from more complex formulas that account for age and height, or encompass mathematical coefficients for increased accuracy. Absolute percentage errors of different calculations that are reported in ref. 18 are quite substantial. As this holds true for healthy subjects already and nobody knows what the relation of cMVV and mMVV looks like in PALS, this study has a major limitation here. Furthermore, there is evidence that not only FEV1 but also maximum inspiratory flow rate (MIFR) substantially contributes to MVV as shown by Dillard et al. in 1993 (PMID: 8466122). Thus, before cMVV(40) is chosen as a new respiratory parameter of interest in PALS, additional work has to be done (or claimed for, at least) that validates cMVV against mMVV and accounts for both FEV1 and MIFR in a smaller patient sample. I am aware that cMVV is easily applicable to retrospective data sets and, probably most important, would spare PALS to actually undergo measurement of MVV in prospective studies. However, the authors should better have taken the chance to include some prepatory work and use this large body of data for proper statistical deduction of the optimal cMVV "entity" in PALS. In case spirometric data sets also comprise peak inspiratory flow (PIF), which is a synonym of MIFR, it would be interesting to combine FEV1 and PIF as Dillard et al. proposed.
It should be also kept in mind and discussed that cMVV(40) is based on absolute FEV1 which does not account for anthropometric measures, which likely explains why it shows limited correlation with mMVV. To conclude this, I feel that the authors did not adequately consider the complexity of the whole MVV issue, and the straight-forward, but simplyfying choice for cMVV(40) almost make one argue why the authors did not use FEV1 itself to answer the questions they had.
2.
Secondly, the authors fail to deduce or justify the necessity of another respiratory measure in PALS in the introduction section. In clinical terms, respiratory monitoring in ALS is all about early detection of respiratory muscle dysfunction, sensitive prediction of nocturnal hypoventilation, and timely initiation of non-invasive ventilation (NIV) in patients in whom nighttime hypercapnia can either be detected or strongly assumed based on daytime ABG measures such as an elevated base excess (Manera U et al. JNNP 2020). The unanswered question is now, whether cMVV(40) is superior to FVC or any other respiratory parameter to improve or accelerate clinical decision-making. The authors praise cMVV(40), with a properly defined, near-median cut-off value of 80, as suitable to distinguish patient populations with different survival time and time-to-NIV, especially in those with FVC above 80% predicted. This is depicted in Figures 1a and 1b, suggesting that combining FVC and cMVV(40) allows for better dissection of prognostic subgroups. Here, it to could be hypothesized more explicitly that MVV, be it measured or calculated, may be more sensitive than FVC to predict respiratory deterioration. I feel that this is plausible as MVV, other than FVC, is co-determined by respiratory muscle endurance rather than strength.
Minor points:
1. Figure 1b: There is an unexpected finding as the graphs suggests that time to NIV is shorter in PALS with FVC<80/cMVV>80 than in PALS with FVC<80/cMVV<80.
2. Page 2, line 54: "MVV is not generally included in the set of lung function tests necessary for diagnosis or follow-up of the pulmonary diseases, because of its good correlation with forced expiratory ventilation in the first second (FEV1), but its role in some conditions, such as neuromuscular disorders, has been reported [15]." The role in neuromuscular conditions should be specified in more detail here.
3. Page 2, line 57: "In clinical practice, a calculated MVV (cMVV) is usually estimated by multiplying FEV1 by a constant value: MVV and cMVV, when compared in some specific populations [16–18]." The term "some speficic populations" is much to vague here and has little to do with the given references: Ref. 16 is about pediatric patients (i. e. unsuitable as a reference for this manuscript), ref. 17 is an unsuitable reference as it gives the advice not to calculate MVV already in its title, and ref. 18 remains as the only adequate citation here. Please revise both sentence and references.
4.. Page 2, line 79: "This dataset reported also the nearest ALSFRS-R performed and we derived from it data on respiratory symptoms (by using ALSFRS-R item 10 and 11) and bulbar dysfunction, defining bulbar involvement as items (1, 2, 3) < 12 [2]." Please make clear that "<12" refers to the Item 1-2 sum score.
5. Page 2, line 81: "Patients with severe pulmonary, metabolic, and kidney diseases and those with signs of uncompensated acidosis/alkalosis (ABGs pH <7.35 and >7.45) were." Please add "excluded" at the very end of the sentence.
6. Page 4, line 136: At diagnosis, cMVV(40) significantly correlated with FVC% (0.626, p<0.001), FEV1% 136 (0.669%), age at onset (-0.426, p<0.001), age at PFT (-429, p<0.001)." In the bracket following PFT, it is likely to change the number into -0.429. Please correct as appropriate.
Reviewer 2 Report
Comments and Suggestions for Authors
The study question isn't novel and useful, lots of clinical measures and methods are better and easier than authors proposal, the clinical used value of this study isn't confident. Therefore, I don't recommend to publish this paper in your journal.
Author Response
Since no specific feedback was provided, all implemented changes referred to feedback by other reviewers.
Reviewer 3 Report
Comments and Suggestions for Authors
In this work, authors propose cMVV as an early marker of respiratory failure based on a retrospective analysis of a large ALS cohort. Although MVV impairment is not new in ALS research, the calculated form as a predictor of time to NIV in patients with a good performance in FVC is interesting. Notably, they analyse a large cohort of patients with robust results. However there are some questions that authors should address:
1- Why authors think cMVV defect precedes FVC? Are the motoneurons or muscle cells involved in cMVV maintenance more vulnerable?
2- As a candidate prognostic biomarker, authors should perform a ROC analysis to estimate a threshold value, sensitivity and specificity for predicting a reasonable time to NIV and fast/normal progression.
3- Authors suggest in conclusions that cMVV assessment can orient clinicians and help with the adherence to NIV. Is cMVV a measure to take into account to start NIV? How can improve the current criteria?
4- As ALS progression follows an anatomical spread, is there any correlation with the site of onset (arms, legs...) in the case of spinal ALS?
5- Could smoking influence in the results? Did authors take it into account in data analysis?
Reviewer 4 Report
Comments and Suggestions for Authors
In the present paper, the authors demonstrated the usefulness of cMVV(40), calculated on the basis of FEV1 from standard respiratory function tests, as an alternative to MVV, as a marker of respiratory failure in ALS patients, especially those with mild/no respiratory dysfunction. The reviewer found this a very attractive paper.
The reviewer has the following concerns, which we would be grateful if the authors could confirm these and amend accordingly:
1. "Thirdly," is duplicated in line 50.
2. the rationale for cMVV (40) was not clearly understood. Please provide a clear description based on the contents of the references cited.
